# Monte Carlo Models of Comet Dust Tails Observed from the Ground

Fernando Moreno 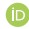

Instituto de Astrofísica de Andalucía, Consejo Superior de Investigaciones Científicas (CSIC), Glorieta de la Astronomia s/n, E-18008 Granada, Spain; fernando@iaa.es; Tel.: +34-958-230603

**Abstract:** Dust particles leaving the comet nucleus surface are entrained by the gas within the first few nuclear radius distances and are subjected to a complex hydrodynamical environment. From distances of about 20 nuclear radii outwards, the particles decouple from the accelerating gas and are mainly affected by solar gravity and radiation pressure for small-sized nuclei. Their motion is then a function of their so-called $\beta$ parameter, which is the ratio of the radiation pressure force to gravity force, and their velocity when the gas drag vanishes. At a given observation time, the position of those particles projected on the sky plane form the coma, tail and trail structures that can be observed from ground-based or space-borne instrumentation. Monte Carlo models, based on the computer simulation of the Keplerian trajectories of a large set of dust particles, provide the best possible approach to extract the dust environment parameters from the observed scattered solar light or thermal emission. In this paper, we describe the Monte Carlo code along with some successful applications of such technique to a number of targets.

**Keywords:** small bodies; comets; main-belt comets; dust; numerical techniques

## 1. Introduction

Comets spend most of their life in the coldest regions of the Solar System, particularly the Oort cloud comets. As a result of this, they are composed of pristine or, at least, minimally-processed materials. Comet dust particles released from the comet nuclei then contain information that can help to constrain the evolutionary processes undergone by such small bodies, such as radial mixing from the innermost hot regions of the early solar nebula to the farthest cold areas beyond Neptune. Evidence of such radial mixing was provided, for instance, by the analysis of comet Wild 2 samples from the Stardust mission, where it was shown that the majority of the particles were produced in high-temperature environments and later transported past the orbit of Neptune, where they accumulated to ice and organic materials to build up the comet nucleus [1]. Current nucleus formation theories suggest that those bodies formed likely through the gentle gravitational collapse of a bound clump of mm-sized pebbles intermixed with microscopic ice particles [2]. Cometary activity can then be explained by ice sublimation and gas diffusion inside those pebbles [3–5]. The Monte Carlo dust tail model is designed to retrieve dust properties from the fitting of comet images. The model has a number of general hypotheses. After leaving the inner coma region, at a certain distance of $\sim$20 R$_N$, where R$_N$ is the nuclear radius, the dust particles decouple from the accelerating gas and are assumed to be subjected to the solar gravity and radiation pressure forces only. The Lorentz force would be important on very small charged dust, and it is neglected based on the arguments given in the following section. On the other hand, radiation forces causing long-term perturbations such as the Poynting-Robertson drag or the Yarkovsky–O'Keefe–Radzievskii–Paddack (YORP) effects are ignored. A complete description of these radiation forces acting on small dust particles can be found in [6]. On the other hand, the nucleus gravity can be neglected in most cases owing to the small nuclei sizes, mainly for short-period comets (their median value is

$R_N$ ~1.6 km, [7]). Under these hypotheses, the motion of each particle is described by the Keplerian orbit around the Sun, and their heliocentric or geocentric positions can be calculated at any time from its orbital elements, which are a function of their velocity and their so-called $\beta$ parameter, the ratio of the gravity to solar pressure force. Monte Carlo codes such as the one that will be described here have been previously used to characterize the dust environments of an ample variety of targets. In this context, it is essential to mention the pioneering work of Marco Fulle, who was the first to develop the so-called Monte Carlo inverse tail model [8], a technique that was successfully applied to a sizable amount of cometary targets observed from the ground (e.g., [9] and references therein).

In the following section, we describe thoroughly the method used to compute the particle orbits and the Monte Carlo procedure aimed at the synthetic dust tail build-up. In Section 3, we show some examples of the performance of the method, and its application to various targets, short- and long-period comets, as well as to an active asteroid, and describe the retrieval of their dust properties. The conclusions are given in Section 4.

## 2. Description of the Monte Carlo Code

### 2.1. Assumption of Particle Sphericity

The Monte Carlo code is aimed at building up synthetic brightness images of comets (or active asteroids) that can be directly compared to dust tail/coma/trail observations. In principle, the procedure is applied to brightness images in a certain observing wavelength, either in the visible or in the infrared domains. The technique uses size distributions of spherical particles only since the calculation of the scattering properties of non-spherical particles for broad size distribution functions would be a formidable task involving huge computational times, as explained in the next subsection. In the backscattering domain, i.e., angular scattering regions near the $0°$ phase angle, and depending on composition, the phase function of non-spherical particles might be very different from that of equal volume spheres [10]. Nevertheless, this problem can be overcome by assuming a certain phase law for the particles, as explained in the next subsections (see Equation (2) below). The Monte Carlo technique could, in principle, also be used to generate synthetic maps of the degree of linear polarization that would provide additional constraints on the dust parameters [11–13]. However, the behavior of the polarization curve as a function of the size and phase angle for non-spherical particles can be a complicated function that is difficult to predict [10].

### 2.2. Particle Physical Properties

We begin with a description of the physical properties of the particles that are needed to develop the Monte Carlo method. The comet dust particles are non-spherical, with a certain structure characterized by a degree of porosity, strength, and a non-homogeneous, composition. Recent reviews of our understanding of comet dust, using both ground-based and space mission data, are given in [14,15]. These works highlight the wide variety of existing particle structures, ranging from compact to extremely fluffy aggregates with sizes up to the mm range, as detected by the Grain Impact Analyzer and Dust Accumulator instrument (GIADA, see [16]) on-board the Rosetta mission to comet 67P/Churyumov-Gerasimenko [17–19], and by the Micro-Imaging Dust Analysis System (MIDAS, see [20]), also on-board Rosetta [21,22]. On the other hand, the Cometary Secondary Ion Mass Analyzer instrument (COSIMA, see [23]), on the same mission, provided, as GIADA, the presence of both fluffy and compact units, the latter consistent with calcium-aluminum-rich inclusions [24]. This leads to a probability distribution of the dust bulk density versus the logarithm of the dust size that is symmetric around the average value of $\rho_p = 785^{+520}_{-115}$ [25,26]. This probabilistic approach has been successfully applied to the dust tail of interstellar comet 2I/Borisov [25]. Future Monte Carlo modeling should, therefore, incorporate this density function since, so far, a single value of the particle density has normally been used for all sizes. In addition, this probabilistic approach considerably reduces the number of model-free parameters. The particle composition naturally affects its refractive index,

which, along with the particle structure, shapes the phase function and, in turn, determines its geometric albedo at zero phase angle. A mixture of silicate, carbonaceous, and organic material is generally assumed for the composition, which, depending on the proportion of those compounds, results in a certain value of the refractive index. As a recent example, Markkanen et al. (2018) [27] considered two populations inside each particle having a highly absorbing component (m = 2 + 0.2*i*) and a non-absorbing, silicate-like component (m = 1.6 + 0.0001*i*).

In the Monte Carlo modeling, the particles are assumed as spheres of radius *r* and have homogeneous composition. The spheres are assumed as independent scatterers, i.e., the particles are located far enough from each other that near-field interactions are negligible, so that each particle scatters radiation as if all the others did not exist. On the other hand, as stated above, the consideration of non-sphericity would lead to excessive complications in the calculations that would imply a huge computing time increment. First of all, the radiation pressure force on a non-spherical particle deviates from the radial direction [28], and it would need a complicated dynamical model, including radiative forces and torques on rotating grains, to describe its trajectory, which will deviate from strictly Keplerian [29]. In addition, for particle sizes larger than or of the order of the observation wavelength, the computation of such forces becomes prohibitive in terms of CPU. For instance, using the Discrete Dipole Approximation [30], to perform such light scattering calculation, a million-sized or more dipole target for a single orientation would be needed, so that for a randomly-oriented target, the calculation becomes impossible in practice on most computing facilities. The calculation of the phase function poses a similar challenge. Then, the only way to overcome those difficulties is to consider spherical particles and to impose a certain geometric albedo at zero phase angle. This albedo has been found to be of the order of 4% at visible wavelengths, consistent with 67P nucleus measurements [31]. Then, instead of a complicated phase function determination, a certain dependence of albedo with the phase angle is assumed, as the classical magnitude–phase relationship adopted for small bodies in the Solar System (e.g., [32]). The linear phase coefficient adopted is then set to be in line with either remote sensing phase function measurements (see, e.g., [33]) or with recent in-situ measurements of the 67P scattering function by Rosetta (see [27,34,35]). The backscattering enhancement of the 67P phase function has been found to be generally in line with that observed from the ground for other comets [34].

Regarding particle sizes, space missions to short-period and Halley's comet have determined the presence of particles from submicron-sized to meter-sized boulders on different comae. Thus, Giotto and Stardust spacecrafts both measured size distributions of particles from nm to mm on comets Halley and Wild 2 [36,37]. Although the cumulative mass power index ($\gamma_M$) varied as a function of the size range, Halley's distribution was, in general, steeper than Stardust ($\gamma_M \sim -1$ versus $\gamma_M \sim -0.6$), so that Halley's comet seems to be enriched in smaller particles with respect to Wild 2. The Giotto extended mission to comet Grigg-Skjellerup revealed a still flatter cumulative mass distribution with an index of $\gamma_M \sim -0.3$ in the *r* > 50 μm range [38], implying a larger depletion of smaller particles with respect to comets Wild 2 and Halley. In any case, inherent to those flyby missions is the simultaneous detection of direct and reflected particles coming from the nucleus, the flux being dominated by the reflected particles, in contrast with the Rosetta observation, where all particles observed come directly from the nucleus [3]. This flyby geometry, therefore, induces an observational bias, absent in the Rosetta measurements. Anyway, the presence of a few small particles in comet Grigg-Skjellerup is in line with the Rosetta/OSIRIS (Optical, Spectroscopic, and Infrared Remote Imaging System, see [39]) findings for 67P. Thus, MIDAS detected a very small amount of micron-sized particles or smaller [14,22]. This is further confirmed by the GIADA microbalance measurements, which are sensitive to the submicrometer- to micrometer-sized dust particles [40]. The small mass detected by that GIADA subsystem implies that the large particle component dominates the optical cross section in the entire 67P orbit. In this regard, VIRTIS-H thermal spectra modeling needed relatively compact particles with minimum sizes of 10 μm particles (combined with

25% fractals in number, [41]). In addition, ground-based modeling of 67P tail, coma, and trail needed particle sizes larger than about 20 μm [42].

Long-period comets could exhibit different dust properties regarding their size distribution, structure, and/or composition. It is well-known, for instance, the significantly higher degree of linear polarization that comet C/1995 O1 Hale-Bopp displayed with respect to all the other comets [43], with the exception of the interstellar comet 2I/Borisov [44]. This might be related to an overabundance of small grains (or, equivalently, very fluffy particles having a high $\beta$ value) as compared to less productive (e.g., short-period) comets. The presence of striae-like features in Hale-Bopp, as well as in other long-period comets such as C/2006 P1 (McNaught) [45], but not in less productive comets, might also be indicative of fragmentation processes [46] leading to an overabundance of small particles in the coma and tail. In this context, and as outlined in the Introduction, the Lorentz force, which will be affecting the dynamical behavior of small charged dust, is neglected in the model. This is based on the following argumentation. The Lorentz acceleration at an assumed heliocentric distance of $r_h$ = 1 au, which would act on the smallest grains of the assumed size distribution, can be written as [45]:

$$a_L = \frac{12\epsilon_0 V_P}{C^2}\beta^2 \frac{\rho_p}{Q_{pr}^2} B_\phi V_{sw} \qquad (1)$$

where $\epsilon_0$ = 8.854 F m$^{-1}$ is the vacuum permittivity, $V_P$ is the potential on the grain surface, generally assumed as $V_P \sim$ +5 V [47,48], $C$ = 5.76 $\times$ 10$^{-4}$ kg m$^{-2}$, $\rho_p$ is the particle density, assumed at $\rho_p$ = 800 kg m$^{-3}$, $Q_{pr}$ is the scattering efficiency for radiation pressure, which takes the value $Q_{pr} \sim$1 from the Mie theory for absorbing particles with $r \gtrsim 1$ (see, e.g., [49], their Figure 5), $B_\phi$ is the azimuthal component of the interplanetary magnetic field, which has a mean value of $B_\phi$ = 3 nT at 1 au from the Sun [50], and $V_{sw}$ is the solar wind speed, which is taken as 400 km s$^{-1}$ [47]. The quantity $\beta$, defined as the ratio of radiation pressure force to gravity force, is given by $\beta = F_{rad}/F_{grav} = C_{pr}Q_{pr}/(2\rho r)$, where $C_{pr}$ = 1.19 $\times$ 10$^{-3}$ kg m$^{-2}$, and $r$ is the grain radius. For the Lorentz acceleration, we then obtain $a_L = 0.0015\beta^2$ m s$^{-2}$. On the other hand, the radiation pressure force is given by $a_{RP} = (1-\beta)GM_{sun}/r_h^2$, where $G = 6.67\times10^{-11}$ m$^3$ kg$^{-1}$ s$^{-2}$ is the gravitational constant, and $M_{sun}$ is the Sun mass. At $r_h$ = 1 au, we have $a_{RP} = 0.006(1-\beta)$ m s$^{-2}$. Therefore, for $r$ = 1 μm particles, we have $\beta$ = 0.74, and hence $a_{RP}$ is larger than $a_L$ in a factor of 2. However, this ratio increases dramatically as $\beta$ decreases. For instance, for $r$ = 10 μm, the ratio is $\sim$700. Since the optical cross section is controlled by particles well in excess of 1 μm, at least for short-period comets, we can confidently ignore the Lorentz force for most purposes.

### 2.3. Particle Ejection Velocities

The initial position and ejection velocity of a particle always refer to the spherical surface of the given radius $R \sim 20R_N$, where it is assumed that the gas drag vanishes. This velocity field must be defined as a function of time or comet heliocentric distance and as a function of size. Those functions can be taken from the results of complex hydrodynamical calculations used to describe the inner coma (see, e.g., [51–54] and references therein). The only available in-situ measurements of particle velocities in a cometary coma are those provided by the GIADA and OSIRIS instruments on Rosetta [17,55]. Della Corte et al. (2015) [17] provided a dependence of velocity on particle mass in the form of $v \propto m^{-0.32\pm0.18}$. On the other hand, simple hydrodynamical considerations [56] provide $v \propto m^{-0.166}$, which is still consistent with the limits imposed by GIADA. Regarding the dependence on heliocentric distance, GIADA also provides estimates from 3.4 to 1.7 au for 67P inbound [18], which we used in the modeling of a 67P dust environment from a large collection of a ground-based data set [57]. In general, given the lack of in situ velocity measurements, the current practice is to assume a parameterization of the velocity as a product of two functions, one giving the time or heliocentric distance dependence and another giving the size dependence (usually taken from simple hydrodynamical models),

in the form of $v(r, r_h) = Cf(r_h)r^{-0.5}$, where $C$ is a constant, and the function $f(r_h)$ must be fitted during the modeling process. Early modeling gave $f(r_h) = r_h^{-0.5}$ [58], which provides a good starting point for the modeling procedure.

### 2.4. Computation of Particle Orbits

The heliocentric position of a given comet in polar coordinates is given by $r_c$, and $\theta_c$, the radius vector from the Sun, and its true anomaly angle, which can be computed at any time from its orbital elements. The comet velocity components, $(V_{rc}, V_{\theta c})$, can be calculated as a function of the specific comet orbit shape (elliptical, parabolic, or hyperbolic) and the true anomaly $\theta_c$. The initial position of a particle is coincident with the comet's position, and its total initial velocity in the heliocentric polar coordinate frame is the sum of its ejection velocity and that of the comet. From the initial position and velocity components in the polar coordinate frame, its heliocentric ecliptic coordinate and velocity vectors can be computed, and its Keplerian orbital path in the Sun's gravity field reduced by the radiation pressure can be calculated from its orbital elements. The acceleration on each particle is given in terms of the $\beta$ parameter defined above.

The heliocentric ecliptic coordinates of the particle are then conveniently transformed into the $(\xi, \eta, \zeta)$ coordinate system (see [59]). In this system, the $\xi$ axis points to the same direction and sense as the Sun-to-comet radius vector, $\eta$ is contained in the comet orbital plane and is directed opposite to its orbital motion, and $\zeta$ is perpendicular to the orbital plane and chosen to form a right-handed set with $\xi$ and $\eta$ axes. The final step is the transformation to the plane-of-sky coordinates, the so-called $(M, N)$ system, where $M$ is the prolonged radius vector from the Sun and $N$ is perpendicular to $M$ and directed opposite to the object's motion along its orbit. To perform that transformation, the coordinates of the Earth in the $(\xi, \eta, \zeta)$ frame at the time of observation must also be computed, for which we need the heliocentric ecliptic coordinates of the Earth. Those coordinates can be obtained from, e.g., the JPL Horizons online Ephemeris System. The transformation of the $(\xi, \eta, \zeta)$ to the $(M, N)$ frame is described in [59].

### 2.5. Computation of Particle Flux

The comet images to be fitted by the Monte Carlo method in the visible spectrum are usually broad band R-filter images. A red band filter is used as they block the blue and UV parts of the spectrum where most of the gaseous emissions take place (see, e.g., [60] and references therein). At the date of observation, if the final position of the particle in the $(M, N)$ system falls within a certain pixel of the image, the magnitude increment, $m$, at that pixel, expressed in mag arcsec$^{-2}$, is given by:

$$p_R \pi r^2 = \frac{2.24 \times 10^{22} \pi r_h^2 \Delta^2 10^{0.4(m_{sun} - m)}}{G(\alpha)} \tag{2}$$

where $r_h$ is the heliocentric comet distance in au, $\Delta$ is the Earth-to-comet distance, and $m_{sun}$ is the apparent solar magnitude in the filter used. For instance, for the Johnson–Cousins R filter, $m_{sun} = -26.76$. The particle's geometric albedo at zero phase angle is given by $p_R$, and $G(\alpha) = 10^{-0.4\alpha\phi}$ is the phase correction, where $\alpha$ is the phase angle, and $\phi$ is the linear phase coefficient. Following Shevchenko's (see [32] magnitude–phase relationship), for $p_R = 0.04$ as given above, $\phi = 0.013 - 0.0104 \ln p_R = 0.046$ mag deg$^{-1}$. For the simulations of IR images, the particle flux can be computed from the equation:

$$F_\lambda = \frac{r^2}{\Delta^2} \epsilon(\lambda, r) \pi B_\lambda [T(r)] \tag{3}$$

where $\lambda$ is the wavelength, $\epsilon$ is the grain emissivity, $B_\lambda$ is the Planck function, and $T(r)$ is the particle equilibrium temperature. According to Kirchhoff's law, the emission and absorption of a particle are described by the same function, $\epsilon(\lambda, r) \equiv Q_{abs}(\lambda, r)$, where $Q_{abs}$

is the absorption efficiency. This parameter can be computed from Mie theory for a given refractive index.

The equilibrium temperature of a single grain can be computed by the balance equation between the absorbed solar and emitted thermal radiation as [61]:

$$\frac{\pi r^2}{r_h^2} \int_0^\infty S(\lambda) Q_{abs}(\lambda, r) d\lambda = 4\pi r^2 \int_0^\infty Q_{abs}(\lambda, r) \pi B_\lambda [T(r)] d\lambda \tag{4}$$

where $S(\lambda)$ is the solar flux at 1 au. The equilibrium temperature can also be approximately computed using:

$$T(r_h) = 278.8(1 - A_B)^{1/4} \frac{1}{\sqrt{r_h}} \tag{5}$$

where $A_B$ is the Bond albedo. If the grain temperature at some heliocentric distance is known from an IR spectrum, the quantity $1 - A_B$ can be estimated, and hence, the equilibrium temperature at any distance. Otherwise, the Bond albedo should be estimated from the assumed phase function $P(\alpha)$ through the equation:

$$A_B = \frac{p_R}{\pi} \int_{4\pi} P(\alpha) \, d\Omega \tag{6}$$

In the Monte Carlo procedure, the ejection of a large number ($\gtrsim 10^7$) of particles are simulated at each time step from the start of cometary activity until the observation time. The size interval considered $[r_{min}, r_{max}]$ is also divided into a certain number of subintervals equally spaced in the logarithm of the radius, and a number of different directions of ejection, according to a certain ejection pattern, are sampled. The ejection pattern might be defined to be spherically symmetric (isotropic) or having an anisotropic character, such as hemispherical or conical ejection (see, e.g., [62]), or any other. Then, the computed total brightness is calculated from the contribution of all particles ejected according to the dust loss rate and the size distribution function, which are both functions of time or, equivalently, the heliocentric distance.

## 3. Application of the Monte Carlo Code to Different Targets

The Monte Carlo code has been applied to describe the dust environment of several small bodies, such as short- and long-period comets, interstellar objects, and active asteroids of various activation mechanisms. The actual version of the code is a parallel program using a Message Passing Interface (MPI) with FORTRAN language. In the following subsections, we explain the way the code has been applied to a sample of those objects, the assumptions made, and the relevant conclusions.

### 3.1. A Conspicuous Short-Period Comet: 67P/Churyumov-Gerasimenko

Comet 67P was selected as the target of ESA's successful Rosetta mission as an alternative comet to the original target, 46P/Wirtanen, owing to a rocket failure. Thus, the launch of Rosetta, foreseen for 2003, had to be postponed to 2004. Being the target of Rosetta, 67P is, undoubtedly, the best-studied short-period comet. Thus, the comet was extensively observed from the ground right after its 2002 perihelion, and sampling two orbits with perihelia in 2009 and 2015 (Rosetta orbit). During the 2002 orbit, the first observation was made by Ishiguro (2008) [63], on 9 September 2002, when the comet was +22 days after its perihelion passage, and on two dates later, on 2 December 2002, and 1 February 2003. The comet dust trail was first detected by Sykes and Walker (1992) [64] from the IRAS satellite, and the neck-line structures become well detected in Ishiguro's observations. The dust trail is the structure that originates from the ejection of large particles, in excess of 1 mm, which are ejected with low relative velocity with respect to the nucleus and are minimally influenced by radiation pressure so that they remain in the nucleus vicinity for a long time within approximately the same orbital path. The neck-line is caused by the ejection of particles, in the current orbit, at the true anomaly of $\pi$ radians relative to the

observation date [65], and becomes prominent when the Earth is close to the orbital plane. This structure is best seen always on post-perihelion observations, as is the case for 67P. Trail imaging when the comet was near aphelion in 2004–2006 was performed, among others, by Tubiana et al. (2008) [66], Kelley et al. (2008) [67] (using the Spitzer telescope), and Agarwal et al. (2010) [68]. Early models of the neck-line were performed by Fulle et al. (2004) [69] and Moreno et al. (2004) [70] from post-perihelion images acquired in March, 2003. The most in-depth analysis pre-Rosetta was provided by the GIADA dust environment model, which used all the available visible and thermal IR imaging data from three perihelia passages, and analyzed the observations using three independent dust tail/trail codes built up by J. Agarwal, M. Fulle, and the one described here (see [71]). Figure 1 displays a comparison of the output of the three models for a 67P image obtained at the 3.6 m Telescopio Nazionale Galileo (TNG) at La Palma. Particle velocities were assumed to be proportional to $\sqrt{\beta}$, and the trail width constrained the dust speed at perihelion. A dust velocity function symmetric with respect to perihelion was found to be consistent with all available images. To fit the visible images, the size distribution, assumed to be a power-law function, was characterized, for $\beta > 5 \times 10^{-4}$, by a power index that is necessarily dependent on the heliocentric distance, varying from –3 in the pre-perihelion branch, and decreasing to $-3.6$ near perihelion, and decreasing even further, to $-4.2$ post-perihelion, i.e., a significant fraction of the mass ejected after perihelion is released in the form of submicron grains. Fits to the trail brightness impose a constraint on the size distribution, which necessarily has a knee at $\beta = 5 \times 10^{-4}$, so that for $\beta < 5 \times 10^{-4}$, the power index is constant to $-4$. This dependence of the size distribution is related to the nucleus pole axis orientation, with the Northern hemisphere, illuminated pre-perihelion, being populated by much larger grains than the Southern one, which is gradually exposed to the Sun after the perihelion. An important conclusion regarding the modeling during the three apparitions is that the comet replicates its dust activity during each orbit, with maximum dust loss rates of the order of 500 kg s$^{-1}$ (for a 4% albedo).

We later performed a systematic analysis with the Monte Carlo dust tail model using OSIRIS images acquired during the Rosetta approach to the target, in combination with ground-based VLT images from 4.5 to 2.9 au inbound [42], and afterward, by monitoring a large portion of the orbital arc from about 4.5 au pre-perihelion through 3.0 au post-perihelion, in combination with the trail data [57]. Sample images acquired during the Rosetta 2015 perihelion are shown in Figures 2 and 3. These images were acquired from the 3.5 m telescope in Calar Alto Observatory on 17 August 2015, close to 67P perihelion, and using the Hyper Suprime-Cam (HSC) on the 8.2-m Subaru telescope on 8 March 2016, at a heliocentric distance of 2.53 au post-perihelion. In this image, both the trail and the neck-line can be clearly seen. Figure 4 shows a comparison of ground-based images with the Monte Carlo model, while Figure 5 depicts the dependence of the dust loss rate with heliocentric distance, together with the water loss rate from ROSINA measurements. The onset of activity was at 4.5 au pre-perihelion or even earlier, and an outburst at 4.1 au inbound was reproduced in the model. The size distribution was dependent on the nucleus seasons, in agreement with the pre-Rosetta analysis by Fulle et al. (2010) [71]. This model incorporated the dust environment results from Rosetta instruments, such as OSIRIS and GIADA, from which the particle velocity distribution was taken. The dust loss rate at perihelion was significantly higher ($\sim$3000 kg s$^{-1}$) than inferred in [71], but still significantly smaller than the mass loss rate derived from OSIRIS data (5000–8000 kg s$^{-1}$). This is related to the fact that decimeter-sized chunks are being ejected from the nucleus [72] that are too big to be actually observed in tails and trails [3].

Dust models comparison

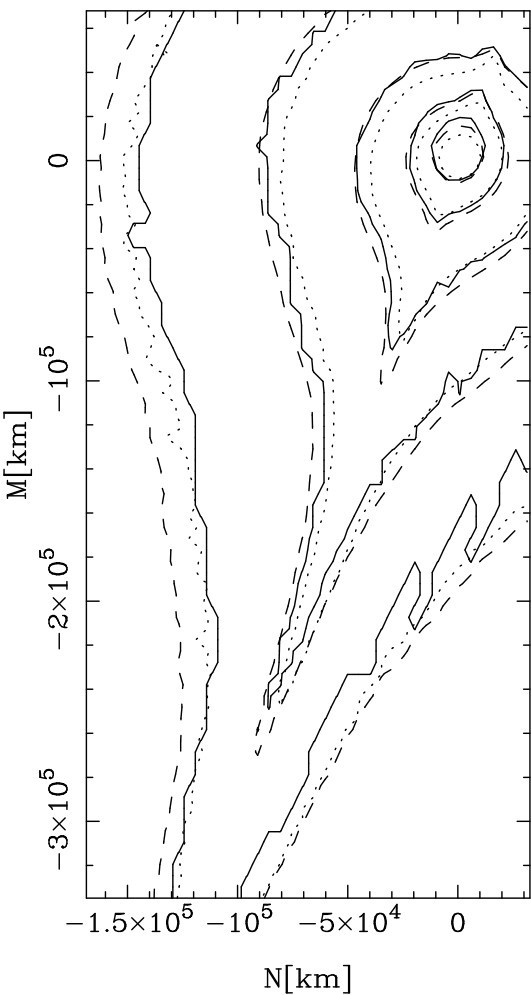

**Figure 1.** Isophote contours of the Trieste (M. Fulle, solid lines), Granada (F. Moreno, dotted lines) and trail (J. Agarwal, dashed lines) codes, modeling the 67P TNG image acquired on 27 March 2003, at $r_h$ = 2.6 AU post-perihelion with isotropic dust ejection. Adopted from [71].

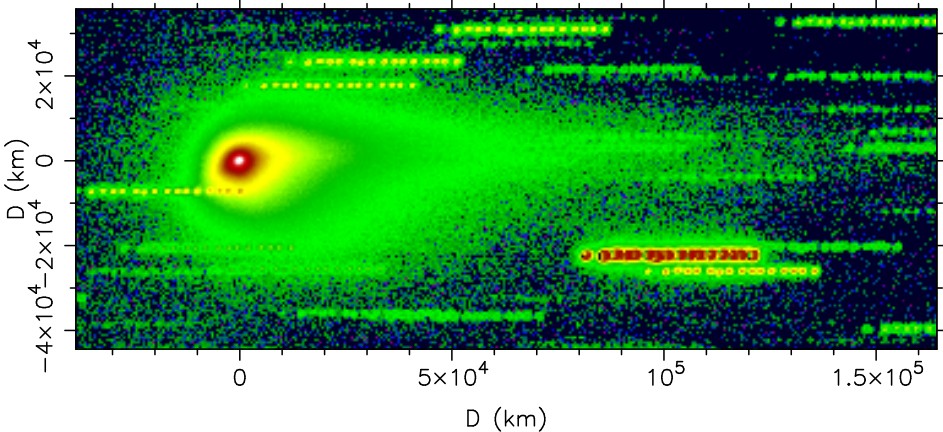

**Figure 2.** Image of comet 67P acquired at the Calar Alto Observatory 3.5 m telescope, using the Multi Object Spectrograph for Calar Alto (MOSCA), on 17 August 2015, at a heliocentric distance of 1.24 au. North is up, east is to the left. Distances (*D*) are projected distances at the nucleus position in the sky. Adopted from [57].

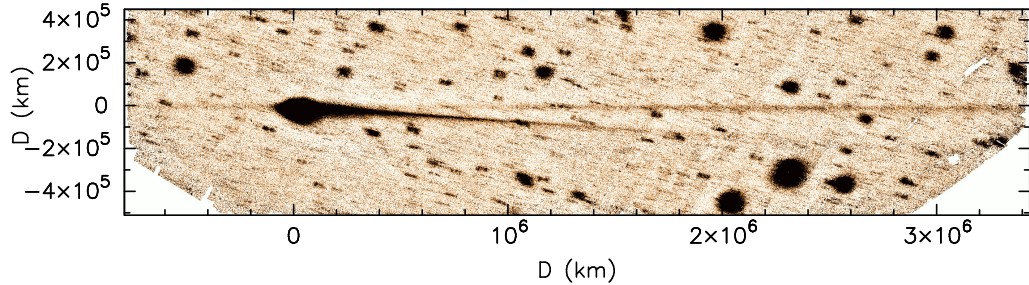

**Figure 3.** Trail and neck-line image of comet 67P obtained on 8 March 2016, at a heliocentric distance of 2.53 au, using the Hyper Suprime-Cam (HSC) on the 8.2-m Subaru telescope at Mauna Kea. The image is oriented so that the trail appears horizontally. Distances (*D*) are projected distances at the nucleus position in the sky. Adapted from [57].

**Figure 4.** Isophote fields of a subset of observed (black contours) and modeled images (red contours) at different heliocentric distances (in au), as follows: (**a**) −4.33; (**b**) −3.80; (**c**) −3.54; (**d**) −3.20; (**e**) −2.95; (**f**) −1.51; (**g**) −1.42; (**h**) −1.37; (**i**) −1.27; (**j**) −1.25; (**k**) +1.25; (**l**) +1.30; (**m**) +1.47; (**n**) +1.62; (**o**) +2.69; (**p**) +3.14. Negative distances indicate pre-perihelion, positive post-perihelion branch. The x- and y-axes are all labelled in km projected at the comet distance. All images are shown in the conventional north-up and east-to-the-left orientation. Adapted from [57].

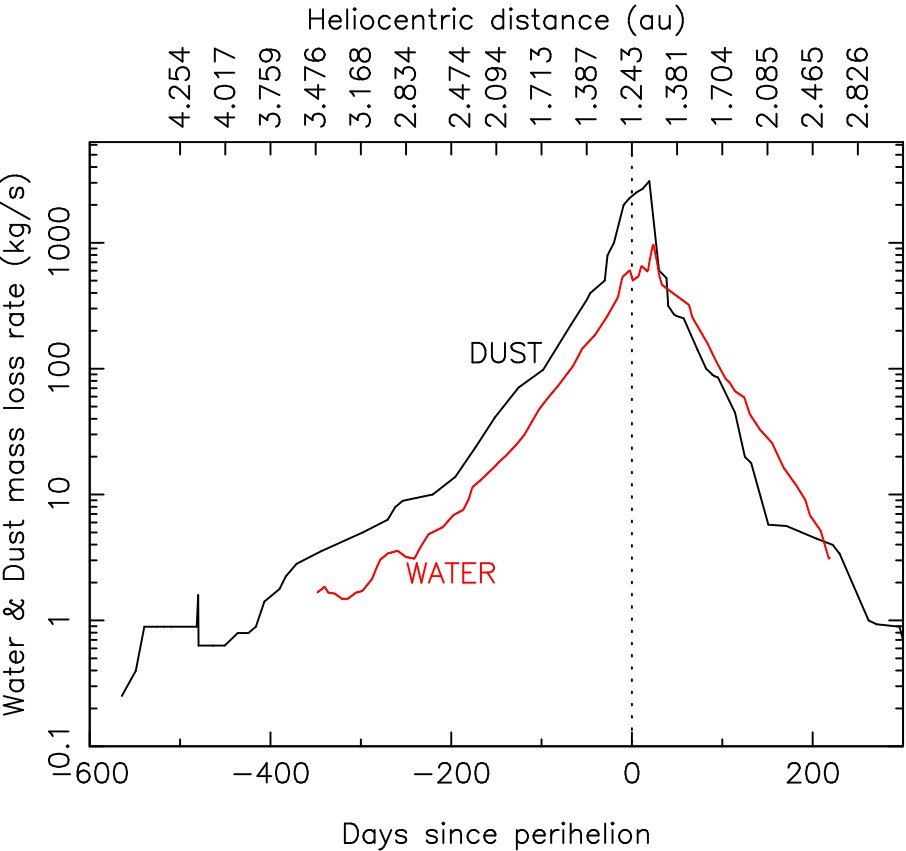

**Figure 5.** Modeled dust loss rate (black line) and water production rate (red line) from the ROSINA instrument on board Rosetta [73], as a function of the heliocentric distance. The sudden increase in dust loss rate at 470 d pre-perihelion (4.11 au) for the modeled dust loss rate corresponds to the outburst observed by the OSIRIS camera on 30 April 2014 [42,74]. Adapted from [57].

### 3.2. Another Peculiar Short-Period Comet: 249P/LINEAR

With a large eccentricity of $e$ = 0.82 and a short perihelion distance of $q$ = 0.5 au, comet 249P moves on a Near Earth Asteroid (NEA)-like dynamically stable orbit. Despite its short perihelion distance, this comet displays only a very moderate activity concentrated near the perihelion. We have analyzed visible images acquired by two perihelion approaches (2006 and 2016) [75]. The Monte Carlo dust tail code is especially well-suited when applied to targets displaying short-term activity, as in this case, as the number of free parameters is considerably reduced. In this case, we performed a fit of the obtained images using only four fitting parameters. The dust loss rate is modeled as a gaussian-shaped function, with a certain peak loss rate $(dM/dt)_0$, full-width at half-maximum (FWHM), and time of maximum ejection $t_0$. The velocity is assumed time-independent and given by as $v = v_0 \sqrt{\beta}$, where $v_0$ is the remaining fourth parameter. The size distribution is set to a broad function with a typical power-law of index $-3.5$ with minimum and maximum particle radii at 1 $\mu$ and 1 cm, respectively. The procedure to find the best-fit relays on the downhill simplex method [76]. In this way, we simultaneously fit all the available images from the two mentioned apparitions, whose results are shown in Figures 6 and 7. The brightness of the nucleus is also constrained, mainly from the images in which it is seen detached from the tail, which implies a size slightly in excess of $R_N$ = 1 km. The resulting peak loss rate was $\sim$150 kg s$^{-1}$ (for 4% particle geometric albedo), very close to the perihelion, with a duration of FWHM = 20 days only. This activity pattern is identical during the two apparitions. As can be seen in the figures, the agreement between the model and the observations is very remarkable. The 249P physical characteristics, combined with its dynamical evolution, likely constitutes an example of an object that originated in the main asteroid belt, thus physically related to the main-belt comets.

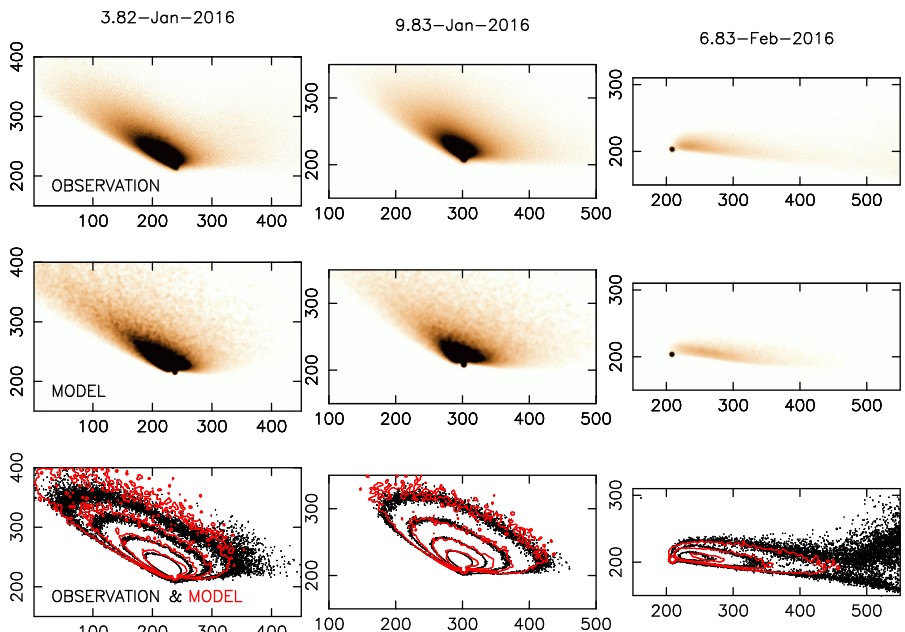

**Figure 6.** Upper panels: observations of comet 249P/LINEAR with the instrument OSIRIS (Optical System for Imaging and Low-Intermediate-Resolution Integrated Spectroscopy) at the 10.4 m Gran Telescopio Canarias (GTC) at the indicated epochs. Middle panels: best-fit synthetic images obtained through minimization in the four-dimensional dust parameter space with the Monte Carlo dust tail code. Lower panels: isophote fields (black contours, observation; red contours, model). All the images are oriented in the north-up east-to-the-left conventional orientation. The x- and y-axis are given in pixels. The physical dimensions of the images are, in km × km, and from left to right, 87,120 × 48,400, 81,000 × 40,500, and 112,080 × 44,832. Adapted from [75].

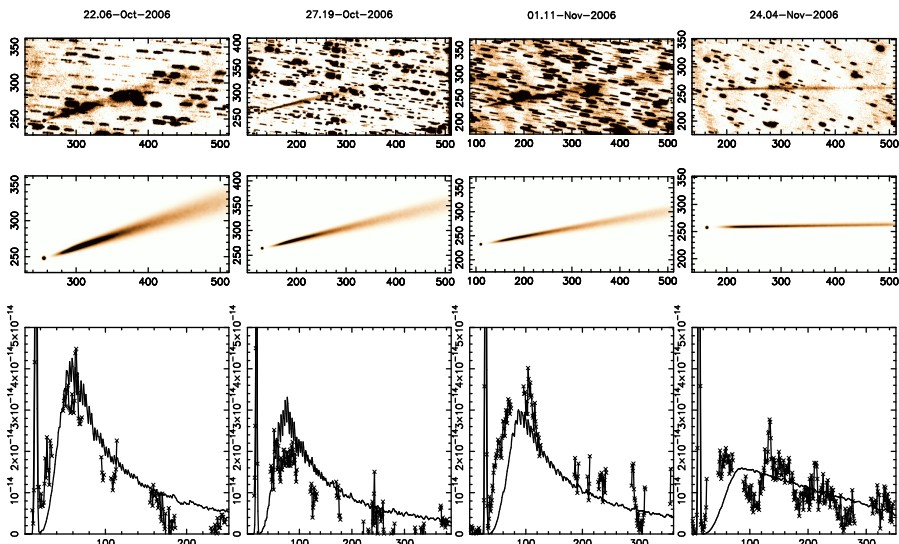

**Figure 7.** Upper panels: observations of comet 249P/LINEAR with the 0.4 m telescope of Great Shefford Observatory near 2006 perihelion at four epochs. Middle panels: best-fit synthetic images obtained with the same dust physical parameters obtained as for the 2016 images in Figure 6 except for a somewhat smaller production rate. The nucleus is seen detached from the tail in the observations and the simulations. The physical dimensions of the images are in km × km and from left to right, 176,734 × 83,683, 266,040 × 126,224, 131,879 × 133,907, and 350,470 × 164,982. Lower panels: scans along the tail for the observed (crosses joined by straight lines) and model (solid lines). The observed scans drawn are limited to the less contaminated regions by background trailed stars. The x- and y-axis of the scans are expressed in pixel and solar disk intensity units, respectively. Adapted from [75].

### 3.3. A Double-Component, Ice-Driven Activity Active Asteroid: P/2016 J1 (PANSTARRS)

Active asteroids represent a new class of small solar system objects that are dynamically main-belt asteroids, yet show transient cometary-like activity [77]. A fraction of the so-called active asteroids are supposedly characterized by ice-driven activity and, as such, are named main-belt comets (MBCs'). Their orbital dynamics in the main belt appear to be long-lasting, and most of them are stable over the age of the planetary system [78]. One of those MBCs' is P/2016 J1 (PANSTARRS), which was discovered as a double-component active object. Orbital dynamics studies predicted a splitting into its two components near its previous perihelion passage in 2010, probably linked to a fragmentation event of their parent body [79]. We monitored the object for several months from May to July 2016 using the GTC (Figure 8). In addition, the object was serendipitously recorded on 17 March 2016 using the 3.6 m Canada–France–Hawaii-Telescope. The two components were separated by some 2 arcmin at discovery time. We applied the dust tail code simultaneously to the two objects assuming a size distribution between $r_{min} = 10$ μm and $r_{max} = 1$ cm, following a power-law function of index $-3.2$. We followed a similar procedure as described in the previous subsection for comet 249P. Thus, we used the multidimensional downhill simplex method in combination with the Monte Carlo dust tail code to retrieve the best fit dust loss rate function (assumed as a half-Gaussian) and the ejection velocity parameters, $v_0$, and $\gamma$, in the velocity parameterization, $v = v_0 \beta^{1/\gamma}$. We found that the two components were activated at the same time, with a long-lasting activity of the order of 6 to 9 months, although with different total dust mass ejected (Figure 9). The ejection velocities of about 0.6–0.9 m s$^{-1}$, only weakly dependent on size, are compatible with escape velocities of 500 m-sized objects. This scenario is therefore compatible with activity driven by ice sublimation.

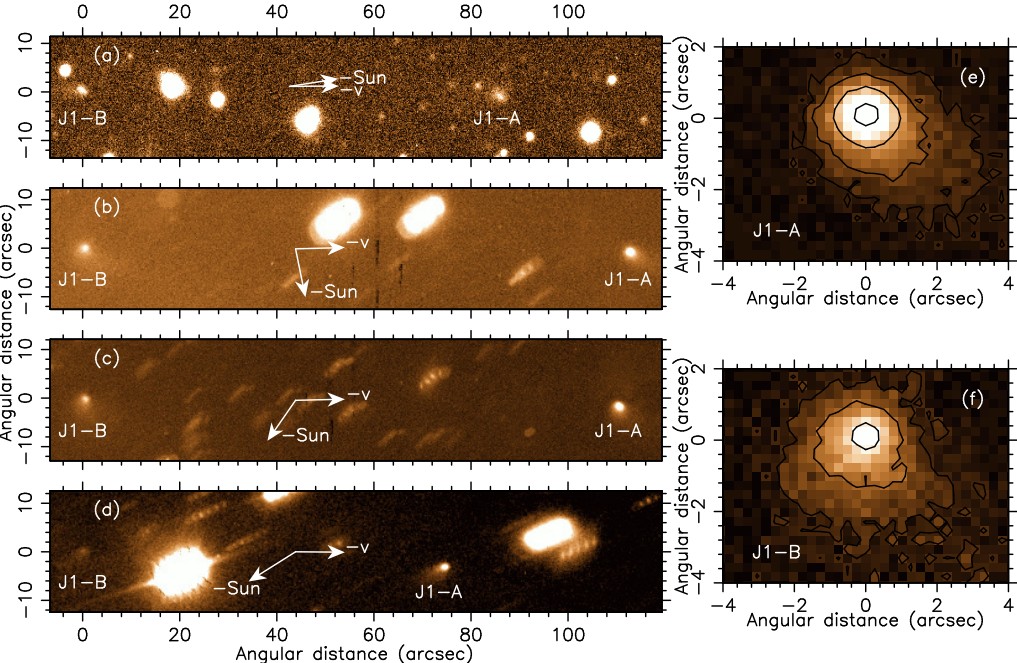

**Figure 8.** Images of the two components, P/2016 J1-A and J1-B, obtained with MegaCam on the 3.6 m Canada–France–Hawaii-Telescope on 17 March 2016 (**a**), and with the instrument OSIRIS at the 10.4 m Gran Telescopio Canarias on 15 May 2016, 29 May 2016, and 31 July 2016 (**b**–**d**). In panels (**a**–**d**), the physical dimensions are 170,982 × 33,925, 133,788 × 26,545, 135,616 × 26,908, and 184,964 × 36,699 km, respectively. Close-up views of J1-A and J1-B on May 15, 2016 are shown on panels (**e**,**f**), respectively. North is up, and east is to the left in all panels. In panels (**a**–**d**), the projected directions opposite to the Sun and the negative of the orbital velocity vectors are shown. Adapted from [79].

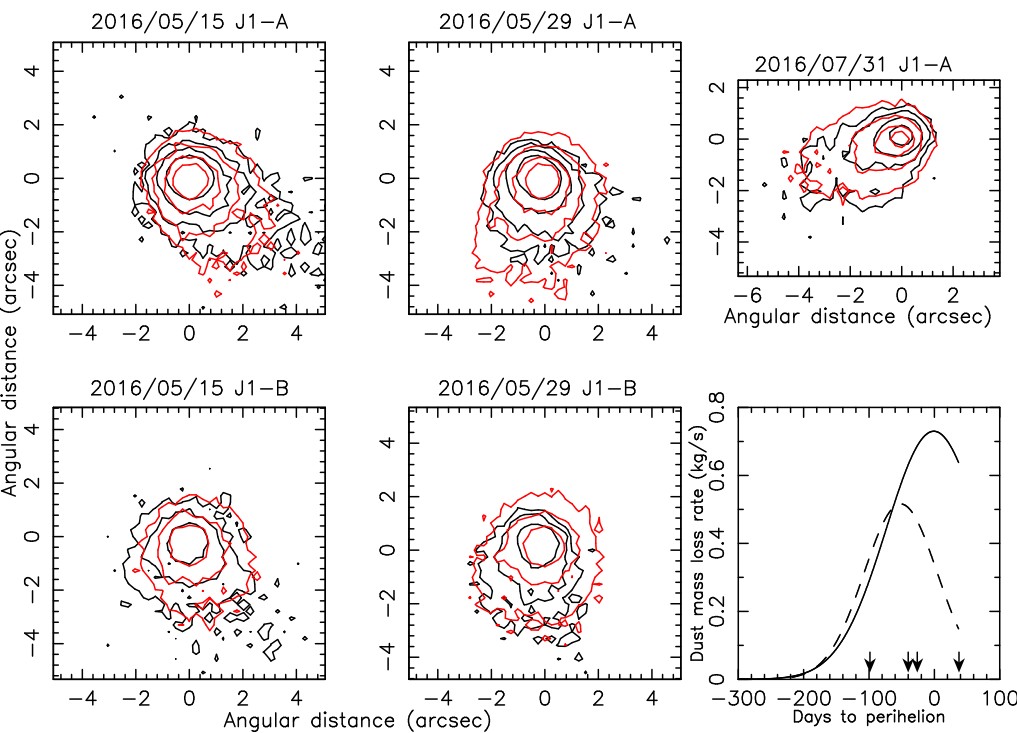

**Figure 9.** Measured isophotes (black contours) and best-fit model isophotes (red contours) for different dates for the two components J1-A and J1-B of asteroid P/2016 J1. The lowermost right panel displays the best-fit dust loss rate as a function of time to the perihelion for components J1-A (solid line) and J1-B (dashed line). Arrows indicate the different observation dates of the images. Adapted from [79].

## 4. Conclusions

The determination of the physical parameters of the dust particles ejected from a comet nucleus from ground-based or space probes is a very complex and challenging task. Apart from the physical properties of the particles, such as size, structure, and composition, the dust environment is a complicated function of the size distribution, the loss rate, and the ejection speeds, which vary with the heliocentric distance. The fitting of the observed brightness of a cometary or asteroidal dust tail involves the assumption of all of those parameters, which are interrelated. Only with dust tail models that take into account all of those parameters is it possible to properly infer the dust environment. For instance, the common practice of conversion of brightness or $Af\rho$ measurements into dust loss rate is nonsense since those observables are a very complex function of all the parameters described above. The problems associated with the Monte Carlo dust tail models come from the large number of intervening parameters. Then, for long-lasting activity objects, such as short- or long-period comets, the more observations available along the object's orbit, the better the dust parameters will be constrained. Furthermore, the dust tail fitting might benefit from independent constraints, such as those provided when in-situ observations are available, e.g., space probes as Rosetta for comet 67P. On the other hand, the activity in main-belt comets, or, in general, in objects displaying episodic activity can also be well characterized using the Monte Carlo models. In those cases, since the activity is confined to a short time interval, multidimensional fitting of the dust parameters can provide a relatively fast procedure to fit the observations, giving important information on the activation times, the dust loss rates, and the ejection speeds, which, in turn, provide clues on the physical mechanism(s) responsible for the activity of that important class of objects. On the other hand, an important implementation based on a probabilistic approach to the particle density distribution on the dust tail models has been recently provided by Cremonese et al. [25]. This procedure limits the number of free parameters in the model

considerably and creates a new scenario in order to retrieve the dust environment of comets and asteroids in the solar system.

**Funding:** This research was founded by the State Agency for Research of the Spanish MCIU through the "Center of Excellence Severo Ochoa" award to the Instituto de Astrofísica de Andalucía (SEV-2017-0709), the Spanish Plan Nacional de Astronomía y Astrofísica through LEONIDAS project RTI2018-095330-B-100, and the Junta de Andalucía P18-RT-1854 project.

**Data Availability Statement:** The observations described in this paper can be accessed through the appropriate archival data system in each observatory.

**Conflicts of Interest:** The author declares no conflict of interest.

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
