# Peer review of "Monte Carlo Models of Comet Dust Tails Observed from the Ground"

_universe, doi:10.3390/universe8070366_

Round 1

Reviewer 1 Report

The paper is a detaield review of dust tail models.

Extensive review of all dust drag processes (included the Lorentz force often neglected) and light scattering are appreciated.

The important implications after the definition of new dust properties provided by the Rosetta mission are taken into account.

A very minor remark: regarding the influence of the huge variability of the dust density on dust tail models, Cremonese et al. 2020 have already proposed a possible solution.

Author Response

I have added the reference of Cremonese et al. (2020) on the probabilistic approach of the beta distribution, that represents an important advance on dust tail modelling, also by reducing the number of dust free parameters. This statement has been included in both section 2.2 and in the Conclusions section, as a future prospect.

Reviewer 2 Report

The aim of this study is to popularize the Monte Carlo code describing the physics of the dust particles that leave the comet nucleus surface, together with some applications of the code. The paper has a good representation of the material and the model is done in parallel FORTRAN. The main thing which remains unclear and I suggest to give more attention to is specification and discussion on the chosen mechanism of particle ejection and their directions, according to the prescribed ejection pattern (isotropic/anisotropic). How are these ejection patterns formed? Based on what theory? Is isotropic here a synonym to "spherical symmetry"? And does anisotropic correspond to directional, conical etc., as in Reach et al. 2010 https://doi.org/10.1016/j.icarus.2010.01.020 and in Gritsevich et al. 2022 https://doi.org/10.1093/mnras/stac822 ?   Could future prospects / remaining knowledge gaps be outlined towards the end of the paper?

Author Response

In order to clarify the referee request on the particle ejection pattern, we have inserted the following statement, making citation of one of the papers that the referee mentions:

"The ejection pattern might be defined as being spherically symmetric (isotropic), or having an anisotropc character, such as hemispherical or conical ejection [see, e.g. Reach et al. 2010], or any other."

On the other hand, a future prospect regarding the incorporation of the density distribution in the modelling procedure, as recently proposed by Cremonese et al. (2020), has been included in the Conclusions section, as well as in section 2.2.